# Fabrication and Characterization of Electrospun Waste Polyethylene Terephthalate Blended with Chitosan: A Potential Single-Use Material

**DOI:** 10.3390/polym15020442

**Published:** 2023-01-13

**Authors:** Thandiwe Crystal Totito, Katri Laatikainen, Chris Bode-Aluko, Omoniyi Pereao, Leslie Petrik

**Affiliations:** 1Environmental and Nanoscience Research Group, Department of Chemistry, University of the Western Cape, Private Bag X17, Bellville 7535, South Africa; 2Department of Separation Science, School of Engineering Science, Lappeenranta-Lahti University of Technology LUT, Yliopistonkatu 34, FIN-53850 Lappeenranta, Finland

**Keywords:** electrospinning, nanofibers, hybrid nanofibers, waste plastic, chitosan, degradation

## Abstract

Textile single-use products are dominantly used for hygiene and personal care, many of which are non-biodegradable and are frequently discarded into sewerage systems, thus causing blockages. Thus, there is a need to move towards water-soluble textiles. This research study focuses on transforming or repurposing biomass material and synthetic reusable waste plastic materials to improve waste. Chitosan (CS) nanofibers could be used in single-use nonwoven fabric or biodegradable tissues, as the water-soluble properties of chitosan nanofibers make them the perfect material for single-use applications. Furthermore, CS was blended with polyethylene terephthalate (PET) polymer and PET-based waste plastic (CS-WPET) to slow the CS nanofibers’ water degradability and strengthen the durability of the nanofiber which could be used as air filters. The CS-TFA and CS-TFA/DCM nanofiber diameters were 95.58 ± 39.28 nm or 907.94 ± 290.18 nm, respectively, as measured from the HRSEM images. The CS-PET and CS-WPET hybrid nanofibers had fiber diameters of 246.13 ± 96.36 or 58.99 ± 20.40 nm, respectively. The thermal durability of the nanofibers was tested by TGA, which showed that CS-TFA/DCM nanofibers had sufficient thermal stability up to 150 °C, making them suitable for filter or fabric use at moderate temperatures. The blended nanofibers (CS-PET and CS-WPET) were thermally stable up to 160 °C. In the aqueous medium stability test, CS-PET and CS-WPET hybrid nanofibers had a slower degradation rate and were easily dissolved, while the CS nanofibers were rapidly and completely dissolved in an aqueous medium. Blending waste PET with CS allows it to be recycled into a useful single-use, non-woven textile, with greater water solubility than unmodified PET nanofibers but more durability than CS nanofibers on their own.

## 1. Introduction

A large proportion of non-degradable plastic waste in the marine environment is the result of single-use synthetic plastics and discarded items [1]. Plastic is used for packaging fresh products in supermarkets all over the world. The medical industry uses plastic in medical equipment such as syringes, instrumentation, tubing, etc., which has revolutionized medicine. The automotive industry is entirely reliant on plastic for dashboards, seat covers, interior surfaces, electric insulation, and bumpers. These are the attractive qualities that have led the world to such overuse of plastic [1,2]. Plastic has great durability and very slow degradability. These materials are used in the production of many single-use products, such as plastic bags, hot and cold beverage cups, plastic water bottles, toys, cutlery, drinking straws, earbuds, wet wipes, etc., that eventually become waste with staying power in the marine environment and on land [2]. The wipes form part of non-degradable plastic waste in the marine environment [3]. Polymer-based PET hygiene wipes have become of concern because they form a big part of single-use products, being a readily available material for a quick cleanup. The popularity of single-use non-woven fabrics such as wipes is then reappearing as part of fatbergs in sewers that are clogged up with grey boulders of congealed sludge stuffed full of wipes [4]. The discarded wipes are also dispersed into rivers and form an insoluble riverbed [5]. Banning single-use fabrics such as hygiene wipes is not likely as there is no announced plan in place to eliminate polymer-based wipes. This problem can be overcome by converting natural organisms’ chitin and chitosan into nonwoven materials or textiles for single-use products such as hygiene wipes, or as disposable filters in the inner linings of masks used for medical purposes. Chitosan is a natural and biologically compatible material that is deacetylated from chitin [6]. The chitin can thus be converted into chitosan polymer and is obtained from the waste of shellfish, the scales of fish, and insects [7]. Chitosan is a biopolymer consisting of natural polysaccharides and is beneficial for hygiene applications because of its special characteristics, such as being antimicrobial and inhibiting the growth of a wide variety of fungi, yeasts, and bacteria [8]. Chitosan is an abundant, inexpensive, biodegradable, nontoxic, biocompatible, renewable, and hydrophilic polymer [6]. Chitosan can bind toxic metal ions, which can be beneficial for use in air cleaning [8]. Chitosan exhibits good reactivity toward a wide range of pollutants [1,6,9]. The use of relatively harmless and earth-abundant local materials is a requirement for the development of any sustainable process, and offers the potential for water-soluble biodegradable textiles. Electrospinning is an accomplished method for producing polymer nanofibers. This technique is competent to fabricate polymer nanofibers at an industrial scale, and it is used mostly in industrial sectors for manufacturing nano-mats for various functionalities. Electrospinning is not only prominent for spinning but for consistently producing nanometer to micron-sized fibers, which are fairly difficult to achieve with other standard mechanical fiber spinning technology methods [10]. This study aimed to produce a moderately soluble, nanofiber-based, nonwoven, single-use textile that would be biodegradable and readily decompose upon being discarded.

## 2. Materials and Methods

### 2.1. Materials

The chitosan (CS) medium molecular weight, polyethylene terephthalate (PET), trifluoroacetic acid (TFA), and dichloromethane (DCM) were purchased from Sigma-Aldrich (Saint Louis, MO, USA) and were used with no further treatment. The waste PET bottles were collected in the Cape Town municipality.

### 2.2. Electrospinning of CS and PET

The electrospinning method of chitosan was adapted from the previously conducted studies with minor modifications [11,12]. The experiment was carried out to optimize the chitosan solvent ratio and composition. The first CS solution was prepared by dissolving 1.12 g of the low molecular weight commercial chitosan with 14.9 g (10 mL) of TFA. This combination made up a solution with a 7 wt% concentration. The 7 wt% CS was dissolved at 60 °C overnight to form a homogeneous solution. The second CS solution was prepared by dissolving 1.09 g of CS in 10.43 g (7 mL) TFA mixed with 3.99 g (3 mL) DCM at a ratio of 70:30%. The low molecular weight commercial CS was dissolved at room temperature overnight to make a homogeneous solution of 7 wt% concentration. Chitosan lysis in TFA produces a more reliable solution for performing electrical radiation, with less interaction between chitosan molecules due to the formation of salts between the TFA and amino groups along the chitosan chain [13]. Each solution was electrospun by introducing 5 mL CS solution into a 20 mL syringe fitted with a 19 G needle, respectively. The syringe was fixed in the pump for controlling the flow rate of the solution during electrospinning. The flow was set at 0.08 mL/hr or 0.1 mL/hr, respectively. The electric field voltage of 25 kV was applied across the needle and the aluminum collector was held at a distance of 12 cm or 14 cm, respectively. The electrospinning process for this experiment lasted for 5 h. The CS nanofibers were collected from the aluminum foil and transferred into wax paper, labeled 7 wt% CS/TFA, and 7 wt% CS/TFA-DCM, respectively.

The PET and WPET solutions were prepared following the same procedures described by Totito et al. (2021) [14]. PET polymer is predominantly used in packaging, and the WPET was obtained as waste plastic water bottles (500 mL) collected from one production batch. The waste plastic water bottles were initially made of a ratio of commercial PET and waste PET during the manufacturing process. The 10 wt% PET or WPET concentration was electrospun by introducing the PET or WPET polymer solution into a 20 mL syringe fitted with a needle. The syringe was fitted into a pump and the solution was electrospun using the following parameters: flow rate of 0.8 mL/hr or 0.05 mL/hr, applied voltage of 17 kV, and a collector distance of 17 cm or 14 cm. The electrospinning lasted for 5 h or 2 h for PET or WPET polymer solution, respectively.

### 2.3. Electrospinning of Blended Polymers

The CS-PET and CS-WPET hybrid nanofibers were produced via an electrospinning process. The volume mixing ratios of the two polymers were optimized in TFA solvent for the electrospinning process. The hybrid polymer solution ratios were prepared by mixing a certain volume of 7 wt% CS and 10 wt% PET or 10 wt% WPET solutions. Amongst the mixing volumes ratios, only a 1:1 ratio was electrospinnable. The volume ratio 1:1 was mixed at room temperature to form a homogeneous hybrid solution. Each polymer solution was electrospun following the same procedure as mentioned in Section 2.2. The syringe was fixed in a pump for monitoring the flow rate of the solution during electrospinning which was set at 0.08 mL/hr or 0.1 mL/hr, an applied voltage of 25 kV or 20 kV, and with the Al foil collector at a distance of 14 cm.

### 2.4. Solubility Test

The PET and WPET nanofibers solubility test was carried out by adjusting the pH of deionized water to 1, 3, 5, 7, 9, or 11 with 0.1 M NaOH or 0.1 M HNO_3_. Then, 0.01 g of the PET and WPET nanofibers were weighed out. The nanofibers were immersed into the pH-adjusted deionized water and shaken at 190 rpm for 24 h. The nanofibers were then washed with deionized water and air-dried overnight at room temperature. The dried nanofibers were weighed thereafter to check the water stability of the nanofibers. The CS-TFA/DCM, CS-PET, and CS-WPET nanofibers were stabilized in various reagents to alter the diffusivity resistance towards aqueous mediums and decrease CS water absorption capacity. The CS-TFA/DCM, CS-PET, and CS-WPET nanofibers (0.01 g) were immersed in either 10 mL of 3 M of NaOH solution in methanol, 10 mL of saturated K_2_CO_3_ solution for 120 min, or neat ethanol solvent for 120 min; thereafter, the neutralized nanofibers were individually rapidly washed with deionized water, dried at room temperature, weighed and characterized by HRSEM.

### 2.5. Analytical Methods

All the nanofibers were studied for morphology, nanofiber diameter, and thermal decomposition using a high-resolution field emission gun scanning electron microscope (HRFEGSEM), using an (Auriga Gemini FEG SEM) (HRSEM), Perkin Elmer (Waltham, MA, USA) 100 FT-IR spectrometer (ATR-FTIR), and TGA 4000 PerkinElmer thermal analyzer, respectively, as characterization techniques. The HRSEM analysis was performed on electrospun nanofibers coated with carbon using a sputter coating technique with an electric field and argon gas to conduct and prevent the nanofibers from charging before analysis. The Scanning Electron Microscopy analysis produced microscopic images at various high magnification levels to allow a better surface analysis [15]. The ATR FTIR wavelength range was set at 4000 cm^−1^ to 650 cm^−1^ against transmittance with 2.0 resolutions at 64 scanning times. For every analysis, the baseline of the FTIR was corrected using the background spectrum of the blank scan. The FTIR diamond crystal and ATR holder were cleaned with ethanol before every analysis to prevent cross-contamination of the samples. The application of FTIR spectroscopy in analyzing the surface of the material involves the determination of the functional groups and chemical structures of various materials. Generally, infrared spectroscopy provides relevant information on short- and long-range coupling orders caused by other factors such as lattice bonding, electrostatics, and deterioration of the original material structure [16]. The TGA analysis was performed in nitrogen with a flow rate of 20 mL/min, and the heating rate was set at 20 °C/min up to 800 °C. Thermogravimetric analysis is a technique that monitors the amount of change in a substance as a function of temperature or time when a sample is exposed to a controlled temperature program in a controlled atmosphere [17]. TGA performs the proximate analysis to determine the amounts of moisture and volatile matter in the material [18].

## 3. Results and Discussion

### 3.1. Characterization of Electrospun CS, PET, WPET, CS-PET, and CS-WPET Nanofibers

CS-TFA, CS-TFA/DCM, PET, WPET including CS-TFA modified PET, and WPET nanofibers were analyzed using the HRSEM to visualize the surface morphology of the nanofibers. The HRSEM images are each presented with a histogram to demonstrate the nanofibers’ average diameter distribution.

The HRSEM images in Figure 1a show the morphology of the electrospun CS nanofibers fabricated from two different polymer solvent solutions. In image Figure 1a the CS-TFA polymer that was dissolved in trifluoroacetic acid (TFA) and then electrospun yielded an average fiber diameter of 95.58 ± 39.28 nm but shows the formation of beads, which can be due to the repulsive forces between ionic groups within the polymer backbone that arise due to the application of a high electric field during electrospinning, which made it difficult to produce bead free CS-TFA chitosan nanofibers. In image Figure 1b, when dichloromethane (DCM) was added to the CS-TFA solution to improve the formation of the electrospun nanofibers, the HRSEM showed bead-free, smooth nanofibers that possessed an average fiber diameter of 907.94 ± 290.18 nm. The CS-TFA/DCM nanofibers thus had a better nanofiber morphology compared to the CS-TFA. The CS-TFA/DCM nanofibers were thus used for further characterization. The difference in diameter size is much more significant compared to the CS-TFA. The expansion in the CS-TFA/DCM nanofibers’ diameter was related to the lower stretching of the CS-TFA/DCM polymer solution at the applied electric field. This could also have been the result of the matrices introduced to the solution crosslinking with the carbonyl functional groups of the polymer, thus increasing the diameter size of the final product. The best solvent for electrospinning chitosan polymer was found to be a mixture of trifluoroacetic acid and dichloromethane in 7:3 ratios with 7 wt% of CS-TFA/DCM.

The HRSEM images with the nanostructure morphology of PET and WPET nanofibers both showed good smooth, uniform consistency. The WPET nanofiber diameters were slightly larger, with an average diameter of 231.44 ± 43.79 nm than that of commercial PET with an average diameter of 124.62 ± 36.73 nm (Figure 1c,d) [19]. The WPET polymer solution flow rate of 0.05 mL/hr and nanofiber collecting distance of 14 cm were the parameters with the strongest influence in improving the electrospinnability of the polymer solution. The WPET polymer solution was spinnable at 0.05 mL/hr flow rate and a collecting distance of 14 cm, yielding a smooth surface and uniform homogeneous nanofibers. These conditions proved to be the optimum conditions in the electrospinning procedure to prepare PET or WPET nanofiber, respectively.

The 1:1 TFA solvent ratio CS-PET and CS-WPET hybrid nanofibers were characterized with the HRSEM to understand the morphology of the electrospun nanofibers. The HRSEM images of the 1:1 CS-PET and CS-WPET hybrid nanofiber blends show consistent, smooth, homogeneous nanofibers with an average diameter of 246.13 ± 96.36 nm for CS-PET hybrid nanofibers and 58.99 ± 20.40 nm diameter for CS-WPET hybrid nanofibers. The significant reduction in the CS-WPET nanofiber diameter size compared to the CS-PET and non-blended CS and WPET nanofibers (see Figure 1a,d) is due to the mechanical forces taking place during the formation of this composite nanofiber (CS-WPET), which may have resulted from the migration of several non-crosslinked oligomers with high molecular weight, or due to the low diffusivity in the PET polymer [20,21]. Thus, this shows good blending was achieved between the natural and the synthetic polymers. It was also observed that CS-PET and CS-WPET nanofibers show fine interfiber pores calculated to be 27.68 nm and 7.22 nm, respectively. This fine porosity would make these mats suitable for air filtration to remove, for example, virus particles which can be 80–220 nm in size [22]. Although the virus size may be smaller or have a similar size as the CS-PET and CS-WPET interfiber pores, it will be difficult for the virus to penetrate through, as the nanofibers are electrospun in multiple layers.

In the ATR-FTIR results (Figure 2a), the distinctive band of the chitosan is present in the combination of the OH and NH functional groups visible at 3284 cm^−1^. The stretching vibrations at 1671 cm^−1^ correspond to their carbonyl (C = O) functional group. The C = O represents the acetyl groups that were directly related to the backbone conformation of the chitosan [23]. The vibrations at the 1533 cm^−1^ band are ascribed to the combination of N-H bending, C-N, and C-C stretching vibrations [24]. The wavelength region of 1381 to 1315 cm^−1^ corresponds to N-H deformation and C-N stretching vibration. The band at 1068 cm^−1^ confirms the C-O functional group [23,24]. The functional groups of the electrospun commercial CS/TFA-DCM nanofiber are similar to the bands 3429 cm^−1^, 1651 cm^−1^, and 1417 cm^−1^ of the commercial chitosan used in the study reported by Kumirska et al. (2010); thus, the chitosan nanofibers were structurally similar to chitosan [25,26], not chitin.

Thermogravimetric analysis of CS-TFA/DCM nanofibers is demonstrated in Figure 2b, showing the thermal stability of the CS-TFA/DCM nanofibers. The thermal decomposition of the CS-TFA/DCM nanofibers started with a 4.27% weight loss between a temperature range of ~15 °C and ~150 °C that is related to the evaporation of moisture, excluding the TFA and DCM solvents, which are volatile and evaporate quickly in a ventilated environment, making it unlikely for these solvents to be trapped in the nanofibers. Thereafter, between temperatures from ~150 °C up to ~400 °C, there was a rapid thermal decomposition with a 61.52% weight loss that was evident by the endotherm centered at 230 °C in the DGA pattern. The first derivative of the TGA results shows the degradation occurred exactly at 273.36 °C. The chitosan thermal decomposition is related to the degradation of the polymer molecules [27]. The slow weight loss of 21.66% in the temperature range ~400 °C to 900 °C was due to the thermal decomposition endpoint of the chitosan. The thermal properties of the CS-TFA/DCM nanofibers would allow their use over a range of temperatures up to 150 °C.

The high thermal stability of the CS-TFA/DCM nanofibers agrees with the thermal degradation of chitosan experiments reported by de Britto et al. [28]. CS-TFA/DCM nanofibers’ thermal degradation was lower compared to the bulk chitosan thermal degradation that ranges from 290 °C to 315 °C reported by Antoniou et al. [29]. Ziegler-Borowska et al. [30] reported a similar thermal decomposition of chitosan in N_2_ and yielded similar outcomes as this research. El-Hefian et al. [31] also investigated the degradation pattern of chitosan and published similar results; however, El-Hefian et al. attributed 42% of the remaining residue to the formation of an inorganic complex containing C, N, and O.

The PET and WPET nanofibers were further characterized using the ATR-FTIR to understand the functional groups of the nanofibers as shown in Figure 2c. The FTIR spectra for PET and WPET nanofibers demonstrate similar FTIR patterns, with the WPET FTIR peaks more pronounced. Thus, the PET commercial nanofibers and the recycled WPET nanofibers possessed similar functional groups, with the WPET consisting of oligomers as matrix interferences. The spectrum of both PET and WPET in Figure 2c shows a weaker band at 2900 cm^−1^ that corresponds to the asymmetric stretching mode of the C-H bond [32]. The carbonyl stretch C = O ester functional group was shown by a strong band at 1720 cm^−1^. The asymmetric C-C-O is present at 1260 cm^−1^ and the O-C-C stretching at the 1100 cm^−1^ band. The strong peak at 700 cm^−1^ represents the shifted aromatic functional group C-H wag, which has been affected by the presence of the carbonyl functional group [33]. Thermogravimetric analysis of PET and WPET nanofibers is demonstrated in Figure 2d for the thermal stability and decomposition range of the nanofibers. The thermal decomposition of the PET nanofibers under pyrolytic inert conditions such as the N_2_ flow rate of 20 mL/min started with an 8.12% gradual increase in weight from 15 °C to ~380 °C. The 8.12% weight increase in the PET nanofibers is related to the expansion of the PET nanofibers, which is due to an endothermic reaction occurring. The successive thermal degradation of the PET nanofibers begins at ~380 °C and ends at ~450 °C, where a significant 70.40% weight of the PET nanofibers was lost. The depolymerization of the PET nanofibers was occurring at this second stage and formed a carbon black residue [34]. The last stage of thermal decomposition in the range of about 450 °C to 900 °C (8.63% weight) was the loss of the carbon by-products that formed [35].

The first stage of the thermal decomposition for the WPET nanofibers started at about 290 °C to ~350 °C, which may be the result of the evaporation of volatile oligomers in the WPET nanofibers’ matrix. The WPET nanofibers lost about 4.55% of their weight in the first stage. Then, the weight of the WPET nanofibers was constant until degradation occurred from 400 °C to ~460 °C. At this stage, the WPET weight loss was about 72.96%. This significant weight loss was the contribution of WPET oligomers’ migration and the degradation of the ester group chain C-C-O together with the unsaturated chain of the carbonyl functional group [33]. This was followed by the gradual weight loss of 9.10% from ~460 °C to 900 °C in the last stage of thermal decomposition, which was the loss of the carbon by-products that formed during the degradation of the WPET nanofibers [35].

The shape of the TGA profiles of PET and WPET nanofibers was found consistent with the increase in temperature. The first derivative plots corresponding with the TGA plots of the PET and WPET nanofibers pyrolysis at about 380 °C to ~460 °C are shown with the endothermic peak degradation in the first derivative plot at exactly 402 °C or 430.60 °C, respectively. Finally, the PET and WPET nanofibers’ thermal decomposition ended when the reaction ceased at a residue accumulation of ~9%. This corresponds with the thermal degradation investigation of waste polyethylene terephthalate (PET) under inert and oxidative environments studied by Das and Tiwari [36].

The CS-PET and CS-WPET nanofibers were characterized using the ATR-FTIR technique to identify the functional groups of the 1:1 CS-PET and CS-WPET nanofibers (Figure 2e). The FTIR pattern of the CS-PET and CS-WPET hybrid nanofibers have weak, broad bands showing at 3653 cm^−1^ and 2964 cm^−1^ that corresponds to the C-H stretching and the combination of the carboxylic acid COOH and amino acids NH_2_ that indicates the presence of the PET or WPET in the nanofibers. The medium bands at 1792 cm^−1^ and 1718 cm^−1^ correspond to the C = O vibrations of the acetyl ester functional group also present in the PET and WPET structural formula. The medium band at 1020 cm^−1^ corresponds to the O bridge stretching in the chitosan structural formula. Asymmetrical C-H bending of the CH_2_ functional group is present at 1371 cm^−1^ and 1267 cm^−1^ bands, which also proves the presence of the CS molecules [37,38].

Thermogravimetric (TGA) analysis of 1:1 CS-PET and CS-WPET hybrid nanofibers is presented in Figure 2f showing their thermal decomposition profile. The TGA and first derivative plot correspond to the CS-PET and CS-WPET hybrid nanofibers’ degradation pattern. The CS-PET hybrid nanofibers had three degradation phases occurring within the 0 °C to 900 °C range. The CS-PET hybrid nanofibers increased in weight by 2.53% from about 15 °C to 130 °C, which may be due to the expansion of the nanofibers [8]. Secondly, there was a rapid degradation of CS-PET hybrid nanofibers that occurred from about 160 °C to ~420 °C where 61.80% of the CS-PET hybrid nanofibers’ weight was lost, which was evident by the endotherm centered at 295 °C in the DGA pattern. The residue of CS-PET hybrid nanofibers’ thermal degradation occurred from about 420 °C to ~900 °C, where 24.84% weight of the CS-PET hybrid nanofibers was lost due to the decomposition of the organic molecules of the two polymers. The decomposition of the CS-PET hybrid nanofibers was evident by the first derivative peak centered at 380 °C. The constant weight from 420 °C to 900 °C was the carbon residue of the pyrolysis reaction that formed. The thermal decomposition of the CS-WPET hybrid nanofibers started with an insignificant decrease in weight till ~130 °C, due to moisture and volatile loss. Thereafter, the CS-WPET hybrid nanofibers slowly degraded in the second stage from ~160 °C to 420 °C and lost 79.53% CS-WPET hybrid nanofibers’ weight, evident by the endotherm centered at 396 °C in the DGA pattern. The last degradation stage occurred from ~420 °C to ~900 °C, where 21.80% of the CS-WPET hybrid nanofibers’ residual weight was lost. The steady weight loss from 420 °C to 900 °C was the carbon residue formed by the pyrolysis reaction. The CS-WPET nanofibers’ thermal stability was approximately 10% less than that of CS-PET nanofibers.

### 3.2. Nanofibers Solubility

PET, WPET, CS-TFA/DCM, CS-PET, and CS-WPET electrospun nanofibers were studied for pH stability and solubility behavior, and the recoverable materials were characterized using HRSEM and ATR-FTIR techniques. The electrospun nanofibers were tested at different pHs for the mass difference that may occur upon exposure to the solution. This will give an insight into each polymer nanofiber’s performance in acidic, basic, or neutral mediums. This stability test showed the pH-related degradation behavior of PET and WPET nanofibers, respectively [39].

In Figure 3a, the PET nanofibers showed a slight weight loss at pH 1 and 3 and insignificant weight loss at pH 7 and 9. At pH 2 PET nanofibers showed an increase in mass, which could be the result of the adsorption of water at pH 2. The WPET showed no significant weight gain or loss at pH 1, 3, and 9 (Figure 3b). Moreover, WPET nanofibers also showed very little weight loss at pH 11, showing their overall stability at various pHs. The PET and WPET nanofibers showed no weight loss or gain at pH 5, meaning both PET and WPET nanofibers are completely stable at pH 5, and above pH 5 no significant instability is evident over the 24 h of the solubility test, showing that these fibers maintained the typical polymer water durability of PET as expected, despite their nano-scale dimensions.

The CS-TFA and CS-TFA/DCM nanofibers were completely soluble in all pH-adjusted aqueous solutions and thus would be immediately degraded in water when discarded after use. The CS-TFA/DCM nanofibers were stabilized to increase the diffusivity resistance in aqueous mediums and decrease its water solubility compared to CS-TFA nanofibers. The CS-TFA/DCM nanofibers were therefore stabilized by neutralization to check if the CS-TFA/DCM nanofibers could be stable in a wet wipes solution for single-use personal hygiene care. The CS-TFA/DCM nanofibers (0.01 g) were immersed in

10 mL of 3 M of NaOH solution in methanol;K_2_CO_3_ solution and ethanol solvent for 120 min.

Thereafter, the neutralized CS-TFA/DCM nanofibers were individually rapidly washed with deionized water. The CS-TFA/DCM nanofibers that had been stabilized in K_2_CO_3_ previously reported by Haider and Park [40] completely dissolved when washed with deionized water, the same as the CS-TFA/DCM nanofibers stabilized with ethanol previously reported by Correia et al. [13]. Both of these stabilization methods did not produce any CS-TFA/DCM nanofiber residues during and after stabilization; thus, this research disagrees with these proposed stabilization methods for CS-TFA/DCM nanofibers electrospun under these conditions (0.1 mL/hr, 25 kV, 14 cm). The CS-TFA/DCM nanofibers stabilized in 3 M of NaOH solution in methanol visually decreased in size, but the remaining material did not dissolve within the 120 min testing period, nor subsequently when washed with H_2_O. Therefore, CS-TFA/DCM nanofibers stabilized with 3 M of NaOH solution in methanol, resulting in nanofibers with stable residues remaining. Further characterization was carried out on CS-TFA/DCM nanofibers stabilized with 3 M of NaOH solution in methanol.

The HRSEM image shows that the CS-TFA/DCM nanofibers stabilized with 3 M NaOH in methanol lost their fibrous character after the stabilization protocol (Figure 4a). The CS-TFA/DCM nanofibers were damaged and formed irregular structures. The CS-TFA/DCM nanofibers lost 50% weight after stabilization with 3 M NaOH in methanol and washing with deionized water as was proposed by Gu et al. [41]. The stabilized CS-TFA/DCM nanofibers were very hard and brittle, thus breaking very easily and losing their nanofiber integrity. The stabilization thus reduced the solubility of the chitosan nanofibers, extending their utility but losing the fabric’s coherence after the process. Thus, these fibers were slightly more durable than untreated fibers but could degrade easily in water. This research is in agreement with the results reported by Gu et al. [41].

The functional groups of the stabilized chitosan nanofibers are shown in the FTIR spectrum (Figure 4b) above. The stabilized chitosan nanofibers’ FTIR spectrum band intensity has decreased and most of the available peaks in the traditional chitosan have disappeared, showing CS decomposition due to the reaction that took place between chitosan and 3 M NaOH in methanol. The broad weak intensity of the absorption band at 3286 cm^−1^ corresponds to the combination of the OH and NH functional groups, which were also present in the chitosan [23,42]. The weak intensity band at 1565 cm^−1^ corresponds to the ammonium ions that may have remained after the stabilization. The weak band at 1650 cm^−1^ corresponds to the aromatics and amine functional groups of chitosan. The bending vibrations at 1410 cm^−1^ were assigned to the carboxyl functional group of chitosan. The band at 1025 cm^−1^ confirmed the presence of C-O functional groups of chitosan [40,43]. Thus, CS and stabilized CS nanofibers could be suitable for dry applications but would immediately degrade upon contact with water, even after stabilization; thus, although 50% of the fiber was retained after stabilization, the fibers were not intact; therefore, the stabilization procedure did not achieve the desired greater durability in water, but showed that this product would not become a problem in sewerage systems after use, due to its relatively rapid dissolution.

The HRSEM images (Figure 5a) of CS-PET and CS-WPET hybrid nanofibers stabilized in 3 M NaOH in methanol show heterogeneous morphology with 146.63 ± 102.80 nm and 353.11 ± 227.31 nm nanofiber diameters, respectively. This shows that the CS component within the hybrid nanofibers was soluble, yielding inconsistent nanofibers after stabilization, fusing the nanofibers and reducing their length. The PET blended CS product after stabilization still had a fibrous character and was more durable than the CS-based products. The CS-PET and CS-WPET hybrid nanofibers’ stability behavior is shown in Figure 5c. After testing the solubility of CS-PET and CS-WPET hybrid nanofibers in water, the mass loss was 79.85% and 8.93%, respectively. The CS-WPET nanofibers showed significantly better water stability than CS-PET nanofibers; this may be due to the WPET polymer embedding CS polymer inside the CS-WPET nanofibers. This could be caused by the fact that the WPET was previously chemically recycled when manufacturing the waste plastic bottles, which introduced oligomer matrices to WPET, thus forming part of the WPET nanofibers. After stabilizing CS-PET and CS-WPET nanofibers using 3 M NaOH in methanol, fibers lost mass by 51.72% and 57.14%, respectively, when exposed to water. The ATR-FTIR pattern (Figure 5d) of the 3 M NaOH in methanol-stabilized CS-PET and CS-WPET hybrid nanofibers had a strong, broad band at 3291 cm^−1^ that corresponded to the C-H stretching and the combination of the carboxylic acid COOH and amino acids NH_2_ that represented the presence of the PET and WPET in the nanofibers. The medium bands at 1640 cm^−1^ and 1563 cm^−1^ corresponded to the C = O vibrations of the acetyl ester functional group also present in the PET and WPET structural formula. The medium band at 1023 cm^−1^ corresponded to the O bridge stretching in the chitosan structural formula, thus showing that traces of CS remained after treatment. Asymmetrical C-H bending of the CH_2_ functional group was present at the 1411 cm^−1^ band, which also proved the residual presence of the chitosan molecules [37,38]. From the discussion presented here, it is clear that further investigation into the stabilization of chitosan is necessary should the fabrics require greater durability. In this sense, introducing less harsh green methods to stabilize chitosan could contribute to elucidating the durability of the PET-blended electrospun CS-based nanofibers.

## 4. Conclusions

The alarming amount of non-degradable single-use synthetic plastic waste found particularly in the marine environment has motivated scientists to implement new strategies for designing and synthesizing single-use biodegradable fabrics or textiles and reusing plastic waste. Chitosan, being the deacetylated polymer from waste shellfish, scales of fish, and insects, was electrospun into CS-TFA/DCM nanofibers using 0.1 mL/hr, 25 kV, and 14 cm conditions. The best solvent for electrospinning a chitosan polymer was found to be a mixture of trifluoroacetic acid and dichloromethane in 7:3 ratios with 7 wt% of CS-TFA/DCM. CS-TFA/DCM resulted in bead-free nanofiber morphology compared to CS-TFA. The thermal decomposition ranges of CS-TFA/DCM nanofibers started at lower temperatures than CS-PET and CS-WPET nanofibers, but CS-TFA/DCM nanofibers had sufficient stability up to 200 °C, which makes them sufficiently durable at moderate temperatures in dry environments. The blended nanofibers (CS-PET and CS-WPET) were electrospun using the TFA solvent, forming hybrid nanofibers that were thermally stable up to 160 °C, with a slower degradation rate and easily soluble in an aqueous medium. While the CS nanofibers completely dissolve in an aqueous medium, blending waste PET with CS allows it to be recycled into a useful non-woven textile with greater water solubility than unmodified PET nanofibers but more durability than CS nanofibers on their own. Stabilization with 3 M NaOH in methanol did not result in a beneficial outcome, as this procedure resulted in deformed nanofibers. CS nanofibers can be used as dry, single-use, nonwoven fabric for air filters, masks, or personal hygiene in medical applications, as the morphology of the nanofiber non-woven fabric presents fine interfiber pores suitable for trapping viruses and bacteria. If used for infection control, these contaminated fabrics could be thermally combusted through depolymerization in a furnace, forming carbon residue. PET and WPET were robust, and when blended with CS their biodegradability properties were enhanced, also affording possible application as a dry air filter or wipe, as they could be discarded after use without concern as the composite fabric would degrade much more rapidly than PET in the environment. The CS-TFA/DCM nanofibers are much more easily discarded, as these nanofibers will completely decompose and dissolve in water with no waste buildup in the environment. Therefore, CS-TFA/DCM nanofibers can be used for single-use personal hygiene care as a highly water soluble, dry nonwoven textile to replace synthetic polymer nanofibers. For water-durable fabric, the CS-WPET nanofibers made using chitosan and waste plastic bottles (0.1 mL/hr, 20 kV, and 14 cm conditions), offered a product that reuses waste and may have an extended utility before it degrades. The CS-WPET hybrid nanofibers were reported on for the first time in this research. For future work, it is recommended that upscaling of the chitosan and composite nanofibers is performed to replace personal hygiene single-use materials, minimize plastic pollution, and minimize marine and land pollution. This study also recommends that further studies be carried out on the intermediate stabilization of the nanofibers. It is recommended that before use, the CS-WPET nanofiber fabrics to be implemented for hygiene care undergo quality assurance and be tested according to ISO 9001, ISO 179, ISO 22716, and ISO/TC 217.

## Figures and Tables

**Figure 1 polymers-15-00442-f001:**
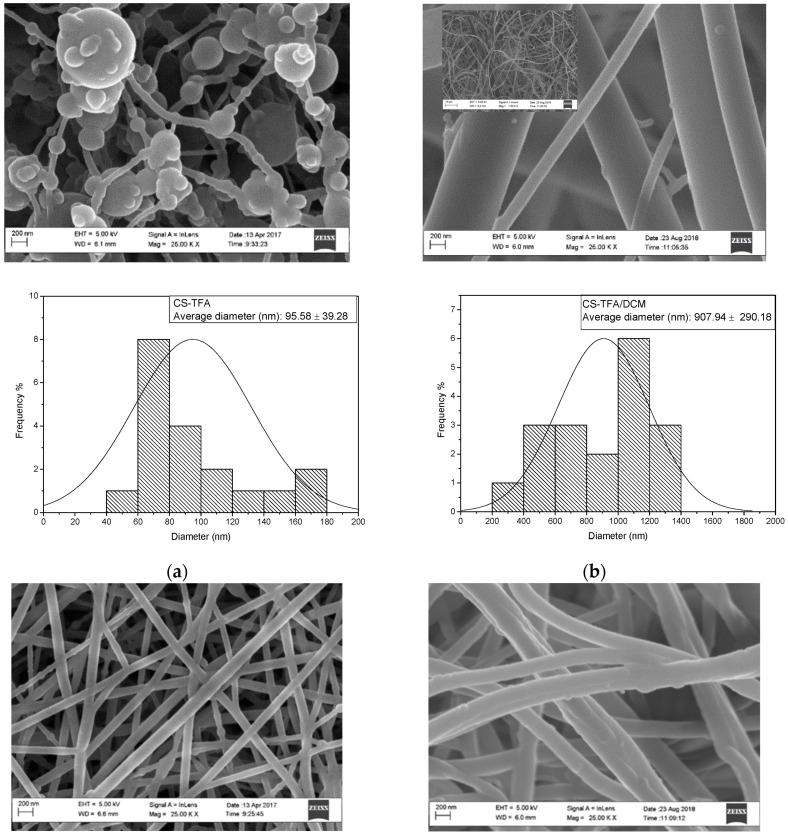
HRSEM images each with the histogram of (**a**): CS-TFA (0.08 mL/hr, 25 kV, 12 cm); (**b**): CS-TFA/DCM (0.1 mL/hr, 25 kV, 14 cm); (**c**): PET (0.8 mL/hr, 17 kV, 17 cm); (**d**): WPET (0.05 mL/hr, 17 kV, 14 cm); (**e**): CS-PET (0.08 mL/hr, 25 kV, 14 cm) and (**f**): CS-WPET (0.1 mL/hr, 20 kV, 14 cm) nanofiber morphology.

**Figure 2 polymers-15-00442-f002:**
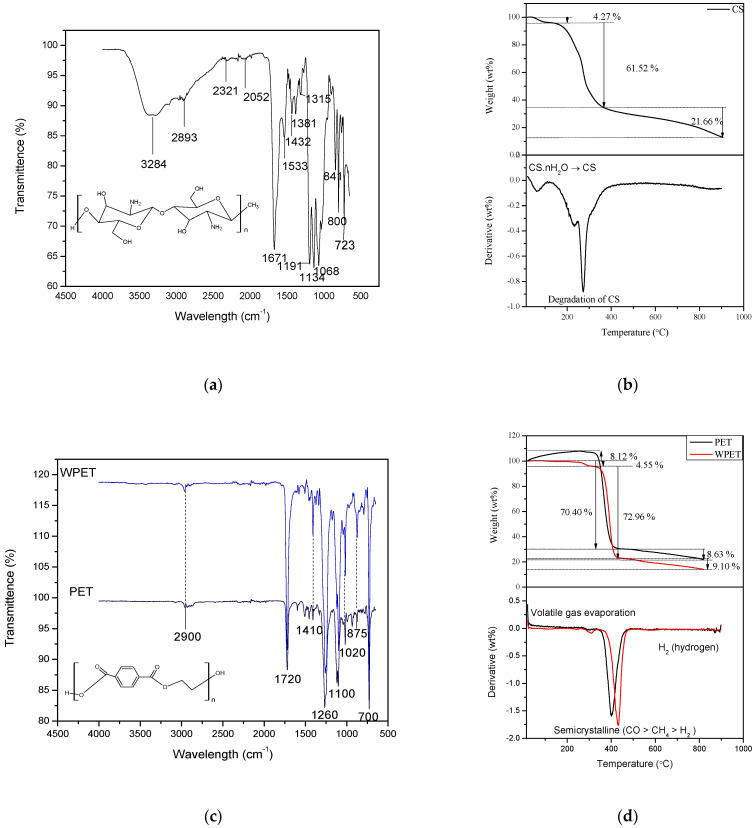
ATR-FTIR pattern of (**a**) CS-TFA/DCM nanofibers (0.1 mL/hr, 25 kV, 14 cm), (**b**) TGA and DGA pattern of CS-TFA/DCM nanofibers (0.1 mL/hr, 25 kV, 14 cm). ATR-FTIR patterns of (**c**) PET (0.8 mL/hr, 17 kV, 17 cm) and WPET (0.05 mL/hr, 17 kV, 14 cm) nanofibers, (**d**) TGA and first derivative pattern of PET and WPET nanofibers (0.8 mL/hr, 17 kV, 17 cm and 0.05 mL/hr, 17 kV, 14 cm), (**e**): ATR-FTIR patterns of CS-PET (0.08 mL/hr, 25 kV, 14 cm) and CS-WPET (0.1 mL/hr, 20 kV, 14 cm) hybrid nanofibers and (**f**): TGA and first derivative pattern of CS-PET (0.08 mL/hr, 25 kV, 14 cm) hybrid nanofibers and CS-WPET (0.1 mL/hr, 20 kV, 14 cm) hybrid nanofibers.

**Figure 3 polymers-15-00442-f003:**
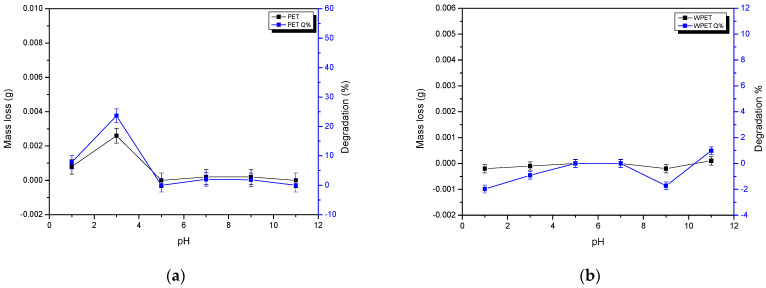
Solubility check of (**a**) PET and (**b**) WPET nanofibers in water at various pHs.

**Figure 4 polymers-15-00442-f004:**
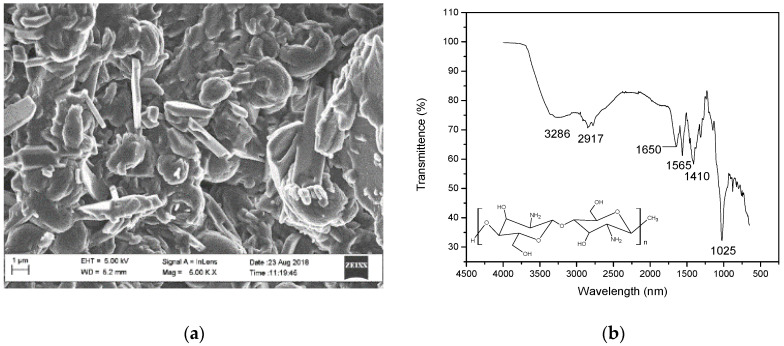
(**a**) HRSEM image of CS-TFA/DCM nanofibers stabilized with 3 M NaOH in methanol and washed with deionized water and (**b**) ATR-FTIR pattern of the CS-TFA/DCM stabilized with 3 M NaOH in methanol.

**Figure 5 polymers-15-00442-f005:**
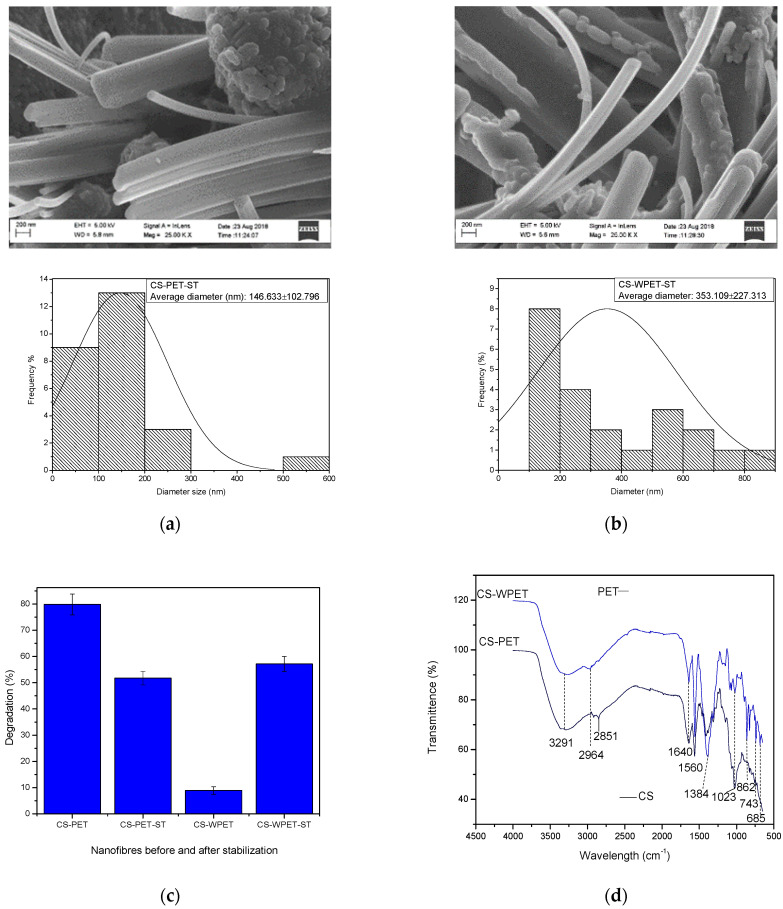
HRSEM images each with the histogram of (**a**): CS-PET and (**b**): CS-WPET hybrid nanofibers stabilized with 3 M NaOH in methanol. (**c**) Degradation percentage of CS-PET and CS-WPET nanofibers before and after stabilization (-ST). (**d**) ATR-FTIR patterns of 3 M NaOH in methanol stabilized CS-PET and CS-WPET hybrid nanofibers.

## Data Availability

Not applicable.

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
