# Peer review of "Fabrication and Characterization of Electrospun Waste Polyethylene Terephthalate Blended with Chitosan: A Potential Single-Use Material"

_polymers, 2023, doi:10.3390/polym15020442_

Round 1

Reviewer 1 Report

The study on PET-CS composites is interesting. Before possible publication authors must cross check the manuscript for grammar mistakes.

Furthermore, the mechanical properties of the produced composites are not studied. Must be added in the manuscript.

Proposed Applications domains must be added in conclusion. 

Reviewer 2 Report

In the paper entitled "Fabrication and characterization of electrospun waste polyethylene terephthalate blended with chitosan: A potential single use material", the Authors are studying a very important environmental aspect, which is the use of polymeric materials as hygienic products. Unfortunately, the paper does not address all the aspects fully, hence I recommend major revision.

1.       I am wondering if the Authors considered what is going to happen after the degradation of PET and CS-PET fibers. Is PET going to be released in the form of nanofibers to the environment? Aren’t plastic nanofibers harder to collect and recycle than PET bottles? There is a lot of concern about the effects of microplastic on environment, what about plastic nanofibers?

2.       If CS-PET is going to be used as a hygienic material, its biocompatibility should be checked. I am aware that this would require more time than the usual time for resubmission of revised manuscript, so I just suggest the Authors to describe safety aspects of their work as a future study (mentioning all required tests necessary to be performed – I am sure that there is some ISO related to this type of products).

3.       Even if I agree with the possibility of using CS-PET as a hygienic product, I have issues with CS-WPET. I understand the idea of recycling of waste plastics, but not necessarily for hygienic products. What about other components of WPET, e.g. additives, colorants, plasticizers, etc – are they going to affect biocompatibility of CS-WPET? Also, the presence of other components might interfere with electrospinning process, hence not all PET waste could be used. Also, IR peaks of PET and WPET are not identical (particularly the ratios of hight of individual peaks), and this needs to be extensively discussed (polymer MW, additives, etc).

4.       Is CS-PET a blend or a copolymer? In the text, Authors are using term “blend”, but Figure 1 presents a copolymer (which is not supposed to be formed during the process).

5.       I would suggest the Authors not to describe different types of fibers separately, but rather combine the description. For instance, by analysing the diameters of CS, PET, WPET, CS-PET and CS-WPET in the same paragraphs would make it easier to see differences and similarities of materials.

6.       Both CS-PET and CS-WPET show fine interfibre pores that would make these mats suitable for air filtration to remove for example virus particles” – this sentencje is far-fetched, because the interfibre area was not calculated.

7.       „The CS-WPET nanofibres showed a significantly better water stability than CS-PET nanofibres, this may be due to the WPET polymer imbedding CS polymer inside the CS-WPET nanofibres” – it is not discussed why WPET should behave different than PET. Also, the differences of stability of before and after stabilization (Figure 7a) is not well discussed in the text. For me, the stabilization effect is significant (particularly for CS-PET-ST), but the Authors state in the conclusions that „Stabilization with 3 M NaOH in methanol did not result in a beneficial outcome.”

Minor issues:

1.       Not all abbreviations are fully named when first appeared in the text (e.g. TFA in abstract)

2.       I am not convinced by showing the range of diameters instead as presenting the mean value +- STD or STE. Also, the paper would benefit if the Authors include and discuss histograms showing the distribution of fiber diameters.

3.       Abstract contains too many experimental details.

4.       Both CS and PET are not detailed described – exact MW should be given.

5.       The last two sentences on page 4 are the instructions for Authors.

6.       Figure 2b should be also presented in lower magnification – to allow observing the structure of fiber network.

7.       Numerous grammar errors.

8.       Figure 5 has strange legend, what is the difference between PET and PET Q%?

Round 2

Reviewer 1 Report

Manuscript can be accepted 

Reviewer 2 Report

The authors did not answer to all my questions. I would like them to think about the following issues:

1. I am wondering if the Authors considered what is going to happen after the degradation of PET and CS-PET fibres. Is PET going to be released in the form of nanofibers to the environment? What about the environmental impact of nanofibers. (The Authors state that "the release of the nanofibers to the environment is not recommended", but their paper is about a potential single use hygienic product made of PET nanofibers, which is going to be discarded into sewerage systems).

2. What about other components of WPET, e.g. additives, colorants, plasticizers, etc – are they going to affect biocompatibility of CS-WPET?  The difference between IR signals between PET and WPET confirms that they are not identical - most probably due to the presence of additives in WPET.

3. Once again, Figure 1 is wrong.

4. When asked about interfibre pores, the Authors wrote that "This research also recommends the study and measurement of the interfibre pores of the CS-WPET and CS-PET composite nanofibre". No recommendation is needed. The Authors have SEM images, so the calculation of interfibre pores is easy and should be done.

5. The CS-WPET nanofibres showed a significantly better water stability than CS-PET nanofibres, this may be due to the WPET polymer imbedding CS polymer inside the CS-WPET nanofibres” – it is not discussed why WPET should behaves different than PET. Also, the differences of stability of before and after stabilization (Figure 7a) is not well discussed in the text. For me, the stabilization effect is significant (particularly for CS-PET-ST), but the Authors state in the conclusions that „Stabilization with 3 M NaOH in methanol did not result in a beneficial outcome.” - this issue was not answered by the Authors.

Round 3

Reviewer 2 Report

I would like to thank the Authors for their response. I have still two doubts:

1. The interfibre pores were calculated to be 0.033 nm and 0.024 nm. Is it really possible? 0.033 nm is 0.33 A, and this is even smaller than the atomic radius of hydrogen, which is 0.5 A.

2. I am sorry for not mentioning it earlier, but the average fiber diameter for CS-TFA and CS-TFA/DCM was calculated as 145 +-170 nm and 1423 +-2000 nm, respectively. How the standard deviation could be higher than the average value?
